# Does the Type Matter? Verification of Different Tea Types’ Potential in the Synthesis of SeNPs

**DOI:** 10.3390/antiox11122489

**Published:** 2022-12-18

**Authors:** Aleksandra Sentkowska, Krystyna Pyrzynska

**Affiliations:** 1Heavy Ion Laboratory, University of Warsaw, Pasteura 5, 02-093 Warsaw, Poland; 2Department of Chemistry, University of Warsaw, Pasteura 5, 02-093 Warsaw, Poland

**Keywords:** tea, selenium nanoparticles, selenium, green synthesis

## Abstract

Selenium nanoparticles (SeNPs) are gaining popularity due to their potential biomedical applications. This work describes their green synthesis using various types of tea. Black, green, red and white tea infusions were tested for the content of polyphenolic compounds and antioxidant properties and then used in the synthesis of SeNPs. In each of the syntheses, nanoparticles with dimensions ranging from 3.9 to 12.5 nm, differing in shape and properties, were obtained. All of them were characterized by a very high ability to neutralize hydroxyl radicals, which was about three-times higher than for the tea infusions from which they were obtained. The main inconvenience in obtaining SeNPs was the difficulties with their purification, which should be a further stage in the described research.

## 1. Introduction

Tea extract obtained from *Camelia sinensis* leaves is consumed worldwide due to its pleasant flavour and nutritional value. The leaves are harvested and roasted, reaching different levels of oxidation, changing the flavour of the tea. Based on the manufacturing process, the main types of tea could be classified as green tea (non-oxidized), white tea (lightly oxidized), oolong tea (partially oxidized) and black tea (fully oxidized) [1]. In the Western world, teas are named by their leaf colour, while the East names them by their brew colour. Thus, for instance, the Chinese named Pu-erh tea (traditionally produced in the Chinese Yunnan province through the microbial process using slowly fermented tea) as red, after the reddish colour of its liquor, whereas the Western world derives its name from the black colour of the finished tea leaves [2]. To avoid any confusion, post-fermented tea is referred to as “dark tea” in English.

Tea is gaining interest for both its therapeutic and nutritional effects. A number of studies have shown that tea infusion possesses a wide range of biological activities, including prevention and/or control of atherosclerosis, hypertension, coronary heart disease, diabetes, metabolic syndrome, obesity and cancer as well as antioxidant, antibacterial, antiviral and antifungal activities [3,4,5,6,7]. Antioxidant properties of tea are manifested particularly by its ability to inhibit free radical generation, scavenge free radicals and chelate transition metal ions. The beneficial effect of tea on human health is related mainly to the presence of phenolic compounds [3,8,9,10]. A variety of these components are responsible for the aroma, colour and taste of tea infusions. Thus, tea extracts, due to their rich composition and various biological actions, play an important role in dietary supplements and cosmetology among others. Tea is also used in food packaging production, as polyphenols delay oxidation and prevent microbial proliferation in certain foods [11].

In recent years, selenium nanoparticles (SeNPs) have attracted great attention due to their higher bioactivity and lower toxicity in comparison to inorganic and organic selenium species [12]. Selenium nanoparticles have an important role in many physiological processes, including growth, reproduction and immunomodulation [13]. Their promising medical uses include treatment of cancer, muscular dystrophy, diabetes or liver fibrosis [14]. More and more attention is also paid to the study of their strong antioxidant and antibacterial effects [15,16]. SeNPs have a high level of absorption in regular supplementation compared to selenium. This is such an important factor also from the point of view of developing new feeding strategies for domestic animals. Some of them have already been developed and described in the literature to increase the Se concentration in animal products [17]. It has been proven that the increased content of selenium in meat is important not only because of the increased nutrient properties in the product but also because it has a positive effect on its other properties, such as colour [17]. The potential role of SeNPs in the environment also seems to be enormous. It has been proved that SeNPs are beneficial for the treatment of water and soil contaminated with heavy metals [14]. Therefore, it is not surprising that many methods of their synthesis have been described in the literature.

SeNPs can be obtained by physical, chemical and biological methods. In the physical approach for selenium nanoparticle synthesis, pulsed laser ablation [18], vapour deposition [19], hydrothermal [20] or solvothermal methods [21] are involved. Out of all those mentioned physical methods, the ablation method seems to be the best, due to the lack of contamination with chemical reagents, easy collection of obtained NPs as well as their high stability. The chemical approach is based on the chemical reduction of inorganic selenium forms as the precursors. Ascorbic acid [16], glucose [22], cysteine [16] and other substances have been used as reducing agents. Usually, the reaction is performed in the presence of a stabilizing agent to prevent aggregation of nanoparticles. However, the main disadvantages of chemical methods of SeNP synthesis are their high cost and the fact that their product can contain harmful substances, which negatively affect human health [23]. This is one of the reasons why the popularity of biological synthesis methods using plant extracts, bacteria or fungi has recently been growing. The point is that the degree of control over the size and shape of SeNPs is higher in chemical synthesis; however, both the toxicity and the cost of the overall reaction are minimized in biological methods.

Among various approaches for SeNP synthesis, the methods involving naturally occurring substances in plant extracts are preferred as they could act both as reducing agents and stabilizers [24,25]. That approach requires non-toxic solvents, mild temperatures and application of the reducing agents that are easily accessible, cheap, biodegradable and not harmful to the environment. It also reduces the necessity of microorganism isolation and a final SeNP purification and, thus, greatly facilitates the application of these methods on an industrial scale [26]. Many factors affect the properties of obtained SeNPs, including pH, extract concentration, type and concentration of the used precursor and temperature. In the synthesis of selenium nanoparticles, the main goal is their formation with minimum particle size and maximum stability, as a smaller size can guarantee more powerful antioxidant activities [16]. However, the proposed procedures for green synthesis of SeNPs do not specify what factors determined the selection of a particular plant. Some of them use little-known plants found only locally and only a few reports are available regarding the synthesis of metal or metalloid nanoparticles using tea extract [27,28,29,30,31].

In this study, the synthesis of selenium nanoparticles using the extracts of different kinds of tea was reported as this topic has not been described before. This is quite an innovative approach because all examined tea types, although derived from one *Camelia synesis* plant, are subjected to different processing methods, which, in turn, affects their chemical composition, particularly polyphenolic profile acting as selenium reductants [32]. It was proven that the total phenolic content of green and black tea is similar, but there are differences in the types and concentrations of particular flavonoids due to varying degrees of oxidation during production [33]. Generally, epigallocatechin 3-gallate (EGCG), the major constituent of tea catechins, is found in higher concentrations in white tea than in green tea [34]. It can be assumed that the content of substances that are commonly used in the chemical synthesis of SeNPs, e.g., gallic acid, ascorbic acid or sugars, also changes significantly due to the oxidation process of tea. 

In this work, we focus on characterizing the differences in the polyphenol profile and antioxidant capacity among studied tea infusions as well as examining their potential in the synthesis of SeNPs. To avoid changes in the chemical composition of teas due to different geographical regions of plantations and different methods of cultivation, all teas came from one producer. The obtained SeNPs were characterized using UV-VIS spectroscopy, transmission electron microscopy (TEM), scanning electron microscopy (SEM) and dynamic light scattering (DLS). The antioxidant properties of the examined tea infusions and synthesized SeNPs were evaluated using DPPH and hydroxyl radical scavenging assays. An attempt was made to link the antioxidant capacity of the tea infusion used for the synthesis with the dimensions and shapes of the obtained nanoparticles as well as with their antioxidant activity.

## 2. Materials and Methods

### 2.1. Reagents

Sodium selenite (Na_2_SeO_3_) used for the synthesis of SeNPs, all polyphenolic standards used during calibration curves development in HPLC analysis, as well as other reagents used during antioxidant capacity measurements were purchased from Merck-Sigma (Steinheim, Germany). Acetonitrile (ACN) and methanol (MeOH), which were used as solvents and eluents in HPLC analysis, were also by Merck-Sigma production. Water used at every stage of the experiment was obtained from the Milli-Q system (Millipore, Bedford, MA, USA).

### 2.2. Tea Samples and Infusion Preparation

All teas used in the experiment were bought in a tea house in Warsaw and came from one producer, Basilur (Angoda, Sri Lanka). The teas were ground in a grinder and then 2 g of each was weighed and poured with 100 mL of water at 60 °C. The brewing process was carried out for 30 min. Then, the infusions were filtered and used for further analysis.

### 2.3. Chromatographic Analysis of Polyphenols in Tea Extracts

Chromatographic analysis of polyphenolic compounds in tea infusions was performed using a Shimadzu LC system coupled to an 8030 triple-quadrupole mass spectrometer (Shimadzu, Japan). MS system was equipped with an electrospray ion source (ESI) operating in negative ion mode. The chromatographic separation was performed in HILIC mode, using a ZIC-HILIC column (100 × 2.2 mm, 3 µm) from Merck. The mobile phase was delivered at 0.2 mL/min in gradient mode: 0–4 min 98% B, 6–7 min 90% B, 8–8.4 min 80% B, 8.4–12 min 50% B and 13–20 min 98% B, where B is acetonitrile and A—water. Polyphenolic compounds were identified based on the knowledge of their fragmentation patterns and retention times, as described earlier [35].

### 2.4. Synthesis and Characterization of Selenium Nanoparticles

The green synthesis of selenium nanoparticles was based on the reduction of sodium selenite with tea extract. Briefly, 2.5 mL of selenium solution (0.1 mol L^−1^) was placed on a magnetic stirrer and 15 mL of deionized water was added. After that, 7.7 mL of tea infusion was added dropwise. The reaction mixture was intensively stirred for 60 min. 

The UV-Vis absorption spectra in a range of 250–900 nm were recorded during the synthesis to observe the progress of the synthesis. It was performed using a Parkin Elmer spectrophotometer (model Lambda 20) with cuvettes of 1 cm in length.

In this work, HR TEM investigations were conducted on an FEI Talos F200X transmission microscope at 200 kV. The morphology and chemical composition were performed in TEM and SEM modes using high-angle annular dark-field imaging (HAADF). Energy-dispersive X-ray spectroscopy (Super-EDS by FEI) detector was used for mapping element distribution. For these studies, the samples were prepared by spotting on Pacific Grid 300 mesh carbon film.

The size of the obtained nanoparticles was also investigated via dynamic light scattering (DLS). For this purpose, Mastersizer 2000 (Malvern Panalytical, Malvern, UK) was equipped with a wet sample dispersion unit (Hydro 2000 MU). Such instrumentation enables a continuous flow measurement of a particle suspension ranging between 0.01 and 2000 µm. The device was controlled by standard operating procedure, which is a part of the software Malvern SOP. The refractive index of the nanoparticles was set to 1.4 and 1.33 for the dispersant (water was used as a dispersant). Measurement adjustments were 15 s for measurement and background time. For each of the samples, 10 independent measurements were carried out, and the presented results are the average of them.

### 2.5. Antioxidant Activity Measurements

The ability to neutralize free radicals by both tea infusions and selenium nanoparticles was tested by DPPH assay and via determining hydroxyl radical scavenging activity. The OH· scavenging activity was performed according to the procedure described by Smirnoff and Cumbes [36]. Thus, 1 mL of tea extract or SeNP solution was mixed with the reaction mixture, which consisted of 1 mL iron sulfate (1.5 × 10^−3^ mol L^−1^), 0.7 mL hydrogen peroxide (6 × 10^−3^ mol L^−1^) and 0.3 mL sodium salicylate (2 × 10^−2^ mol L^−1^). The sample was then incubated for one hour at 37 °C and, finally, the absorbance was measured at 562 nm. The scavenging activity of the samples was calculated as follows:$$\%OH=\frac{1-\left(A1-A2\right)}{A0}\cdot 100\%$$
where *A*0 is an absorbance of the blank (without sample addition), *A*1 is the absorbance with the sample and *A*2 is the absorbance without sodium salicylate.

In the DPPH assay, 0.1 mL of the sample (tea infusion or SeNP solution) was mixed with 2.4 mL of alcoholic solution of radical (9 × 10^−5^ mol L^−1^). After 30 min, the decrease in the absorbance at 518 nm was measured. The results are expressed as Trolox equivalent (TrE).

Folin-Ciocalteu assay was also performed to monitor how the total phenolic content changes during the synthesis. In this method, 1 mL of the sample was mixed with 0.1 mL of FC reagent and 0.9 mL of water. After 5 min, 1 mL of 7% solution of Na_2_CO_3_ and 0.4 mL of water was added. After 10 min, the absorbance was measured at 765 nm. The results were expressed as gallic acid (GA) equivalent.

For CUPRAC method, the procedure described by Apak et al. was used [37]. In this method, 1 mL of CuCl_2_ solution (1.0 × 10^−2^ mol L^−1^) was mixed with 1 mL of neocuproine solution (7.5 × 10^−3^ mol L^−1^) and 1 mL of 1 mol L^−1^ NH_4_AC buffer (pH 7), followed by mixing 0.5 mL of extract and 0.6 mL of water. After 20 min of incubation at 50 °C, the absorbance was measured at 450 nm. The antioxidant activity of the extracts was expressed as Trolox equivalent (TrE).

### 2.6. Statistical Analysis

The experimental results from antioxidant activity measurements as well as HPLC analysis were obtained from at least three parallel measurements and are presented as average ± SD. The significance of differences among means was carried out at a 5% probability level using one-way ANOVA and Tukey’s test.

## 3. Results

### 3.1. Characterization of Teas Used for SeNP Synthesis

The absorbance spectra of studied tea infusions were recorded in a range of 200–850 nm, which is shown in Figure 1. According to the literature, such spectra can be the basis for distinguishing different kinds of tea [38,39,40]. Absorbance in the region from 250 to 350 nm is related to the η → π electronic transition of methylxanthines and catechins, and specifically, these compounds can be used as chemical descriptors to distinguish black tea from green tea [39]. Other groups of compounds, which can be useful in the interpretation of tea spectra, are theaflavins and thearubigins, typical for black tea as a result of fermentation. Their presence in the infusion can be examined at 380 and 460 nm [40]. As can be seen in our spectra, black tea shows higher absorbance than green tea in the region of 250–400 nm but, on the other hand, the intensity of the absorbance is not significantly different from that obtained for red tea. In a range of 300–400 nm, green tea shows higher absorbance than red tea, which is consistent with literature data [38]. Additionally, green tea infusion presents the lowest absorbance, in the region 450–600 nm, which was also reported before [38].

The quality of tea infusions was also tested in terms of the content of individual polyphenolic compounds. The results of the chromatographic analysis are presented in Table 1. This aspect is also important in terms of the use of teas for the synthesis of SeNPs, as polyphenolic compounds are potential selenium reducing agents and can also act as stabilizers for the obtained nanoparticles.

Teas are known for their antioxidant abilities. Table 2 shows the antioxidant capacity of both tea extracts and SeNPs synthesized with them. In order to compare the obtained results, the antioxidant potential was determined in accordance with the procedure proposed by Seeram [41]. For all assays, appropriate index AOX was calculated as a percentage of the average antioxidant capacities of a given sample compared to the highest one (sample score/best score × 100). Then, AOX index was obtained for each sample as the sum of the individual indexes divided by four (number of tests).

### 3.2. Synthesis of SeNPs

The synthesis of SeNPs was conducted parallel by the reduction of sodium selenite with a tea infusion. The excess of tea to selenium was 1:3. The changes in the UV-Vis spectrum were monitored from the beginning of the synthesis and then every 15 min until the hour, i.e., the end of the reaction. The recorded spectra are presented in Figure 2.

### 3.3. Characterization of SeNPs

The morphology, shape and size of synthesized nanoparticles of selenium were evaluated via SEM and TEM analysis. Transmission electron microscopy is a powerful technique to determine the nature and morphology of very small particles, but in the case of our study, the identification was made difficult by an inhomogeneous environment where the nanoparticles’ background contrasts and varies, depending on the examined sample region. It should be noted that in the case of the described syntheses, problems with the purification of the obtained SeNPs were encountered. The nanoparticles were so strongly stabilized by the sample matrix that they did not precipitate out of the solution either under the influence of centrifugation (regardless of its speed) or their self-aggregation and falling out of the solution after synthesis (6 months). That is the reason why in Figure 3, the quantitative analysis of high-angle annular dark-field scanning TEM (HAADF STEM) images is presented. According to the literature, it can also improve the accuracy of semiautomatic segmentation methods based on morphological processing to calculate size histograms [42]. The relevant histograms are shown in Appendix A.

## 4. Discussion

### 4.1. Polyphenolic Content and Antioxidant Activity of Tea Extracts

The recorded UV-Vis spectra of tea infusions confirmed that they were obtained from different types of tea, i.e., black, green, red and white. It was, therefore, to be assumed that this would translate into the content of polyphenolic compounds in them, which could potentially reduce selenium to the form of nanoparticles. At the same time, they are also responsible for the antioxidant capacity of tea infusions. HPLC analysis of studied tea infusions showed that tea extracts are a good source of polyphenolic acids. Gallic, chlorogenic, protocatechuic, p-coumaric acid and pHBA were detected in all studied infusions. Caffeic acid was present in small amounts in all teas, with the exception of green tea. In plants, caffeic acid is formed from p-coumaric acid and then transformed into ferulic acid [43]. However, in the case of the studied teas, the concentration of ferulic acid was below the limit of detection (<0.010 mg L^−1^). The highest levels of p-coumaric and caffeic acid were found in red tea, followed by black. In the relation to catechin content in the analysed infusions, white tea contains the highest level of these compounds, followed by green tea. On the other hand, the highest concentration of EGCG was found in the green tea infusion. It is crucial from the point of view of the antioxidant activity of the samples, as this compound is known to be a strong antioxidant [44]. Rutin, the glycoside combining the flavonol quercetin and the disaccharide rutinose, was present in all studied infusions; the order of its concentration was: black tea > white tea > red tea > green tea. The differences in flavonoid content are the result of the tea variety and its method of cultivation [45]. It should be highlighted that the used conditions have a great impact on the extraction efficiency of polyphenolic compounds. As the aim of the described research was the green synthesis of nanoparticles, aqueous extracts were tested. However, also here, the tea brewing time, the temperature of the water used and the weight of the sample itself may vary depending on the research.

Tea infusions are known and appreciated, among other things, for their antioxidant properties. For the tested extracts, their ability to neutralize DPPH and OH radicals, total polyphenol content (Folin-Ciocalteu assay) and reducing properties (CUPRAC method) was determined. For each method, the total antioxidant index (AOX) was also determined as a percentage of the average antioxidant capacities of a given sample compared to the highest value. The collected results are presented in Table 2. The highest content of polyphenolic compounds was determined for white, followed by green, tea infusion. The corresponding values for black and red tea were significantly below 1000 mg GA/L. White and green tea also showed a similar ability to scavenge the DPPH radicals. It should be mentioned that these two types of tea undergo only a minimal procedure at the production stage, most often drying, hence, the similarities in their properties. With the progress of the fermentation process, the ability to neutralize DPPH radicals decreases and, thus, red tea, which is subjected to a half-fermentation process in relation to the black variety, has lower antioxidant properties in this assay. This trend is reversed compared to the results obtained by the FC method where red tea infusion shows a higher content of polyphenolic compounds. 

A clear trend between the ability of the brew to neutralize free radicals and the degree of tea processing is visible in the results obtained for scavenging hydroxyl radicals. The ability to scavenge OH radicals increases in the order: black tea < red tea < white tea < green tea. On the other hand, there is no such dependence if the results of the study for the reducing capacity of the samples are taken into account. In CUPRAC assay, the results obtained for green and black tea infusion were not statistically different, while red tea possesses the lowest reducing activity from all studied tea varieties. The values of the calculated AOX index increase in the order: black tea < green tea < white tea < red tea, which is also not following the degree of tea processing. However, the antioxidant capacity is affected not only by polyphenolic compounds but also by the presence of other compounds in the sample, for example, some inorganic substances affect the result obtained by FC assay [46].

### 4.2. Synthesis of SeNPs

The progress in the chemical synthesis of SeNPs can usually be observed by colour change. The formation of brown colour in the reaction mixture could confirm the presence of selenium nanoparticles. However, in our reactions, it was impossible due to the colour of the infusion itself, particularly when black and red teas were used. In the case of green and white teas, a change in the colour of the reaction mixture from light green or straw to a characteristic brown-red was also not spectacular. Information can be found in the literature that identifying the brown colour of the solution with the formation of selenium nanoparticles is ambiguous due to the strict dependence of the colour of the solution on the size of the suspended nanoparticles [47]. The authors postulated that the suspension of nanoparticles with a size of approx. 20 nm has a yellow-orange tint, and only as their size increases, the colour of the solution changes to red-brown. Nevertheless, in the case of green SeNP synthesis, it was necessary to monitor the progress of the reaction in a different way than visually. Recording the UV-Vis spectra can be a valuable source of information, as Lin and Chris-Wang related the location of the absorption maxima of selenium nanoparticles to their sizes [48]. According to this simple method, 18.1 ± 6.7 nm-sized SeNPs exhibit an absorption maximum of around 250 nm. As the size of the nanoparticles increases, their absorption maximum is shifted towards higher wavelengths, e.g., SeNPs with dimensions of 48.2 ± 5.9 nm show an absorption maximum of about 250 nm and those of 101.6 ± 9.8 nm around 350 nm.

In our research, the recorded UV-Vis spectra show that both the reaction progress and the resulting SeNP sizes differ depending on the type of used tea infusion. In the case of synthesis involving green tea infusion, only small changes in the spectra are visible in the region around 400 nm after 60 min of synthesis (Figure 2). This may suggest that the synthesis reaction is fast and the final product is received just after the mixing of the reagents. On the other hand, a slight increase in the absorption band around 400 nm may suggest possible aggregation of SeNPs, as this region corresponds to nanoparticles with dimensions between 101 ± 9.8 and 146.1 ± 23 nm. At the same time, however, no decrease in band intensity was observed at lower wavelengths. In the case of the synthesis carried out with the use of white tea, an increase in band intensity was observed at 350 nm, corresponding to nanoparticles with dimensions 70.9 ± 9.1 nm. This trend persists until 45 min of the reaction and then we observe a decrease in the intensity of this band, with a simultaneous increase in the band around 250 nm. This may suggest the formation of nanoparticles smaller than 70 nm only in the final stage of the reaction. Similar relations were obtained for the synthesis with black tea infusion, where an increase in the intensity at 300 and 400 nm was observed. After an hour, their intensity decreases in favour of the band around 250 nm. In the case of the use of red tea, an increase in the intensity of the bands at 300 and 400 nm was observed throughout the reaction. The greatest increase in the intensity of these bands was recorded after the first 5 min of the reaction; thus, SeNPs with dimensions of 70.9–101 nm should be expected.

Based on the determined decrease in the concentration of Se(IV) in the post-reaction mixtures, the yields of individual synthesis reactions were calculated. In all cases, they were higher than 90% and increased in a series: red tea (91.40 ± 3.9%) < white tea (92.16 ± 4.0%) < green tea (92.31 ± 4.1%) < black tea (92.50 ± 4.30%). This order is consistent with the decreasing of AOX index for the tea used for the synthesis. However, it does not correspond to the total content of polyphenol compounds determined for teas using the FC assay. This may suggest that the potential of a given infusion in SeNP synthesis is determined not only by the high content of polyphenolic compounds, i.e., potential selenium reducers, but also by other factors. However, significant decreases in the concentrations of polyphenolic compounds in the post-reaction mixture suggest their main participation in the synthesis of selenium nanoparticles. These data are presented in Appendix A. The concentrations of all flavonoids and polyphenolic acids detected in the extracts significantly decreased, and the concentration of some was below their limit of detection, e.g., catechin. It should be noted that polyphenolic compounds are the main but not the only potential selenium reducers in the tea extract. Tea infusions naturally contain substances that are used as reagents in the chemical synthesis of SeNPs, e.g., ascorbic acid. Its concentration was also determined in the used tea infusions. Its highest level was found in green tea infusion (1.85 ± 0.06 mg/L) followed by white tea (1.31 ± 0.05 mg/L). In the extracts of black and red tea, the concentration was slightly lower and amounted to 1.10 ± 0.04 and 0.902 ± 0.01 mg/L, respectively. After the nanoparticle synthesis, the concentration of ascorbic acid in the reaction mixture was below its limit of detection (0.01 mg/L). This confirms its role in the synthesis of SeNPs.

### 4.3. Characterization of SeNPs in Terms of Size and Morphology

The green synthesis of SeNPs using tea extracts is fundamentally different from other methods described by the authors in their previous work [16,27]. Although in one of the described procedures, extracts from some medicinal plants were used, the colour change during the synthesis was clear and there were no problems with the purification of the obtained nanoparticles [31]. How important a step in the synthesis of SeNPs is their purification, as shown in the example of chemical syntheses using ascorbic acid and cysteine [16]? In the case of synthesis using tea infusions, no colour change is observed, but this can be explained by the small size of the formed SeNPs. According to the literature, the smaller the dimensions of the SeNPs, the more yellow the colour of their suspension [47]. The biggest problem in this approach for SeNP synthesis is the difficulty in their purification. The use of centrifugation does not cause the SeNPs to fall out of the solution. They are sufficiently stabilized by substances from the extracts that their self-aggregation and loss are not observed even after 6 months from the synthesis. Therefore, to study the result of the nanoparticle synthesis, the post-reaction mixture should be examined. However, the nanoparticles form clusters and are covered with other substances, which may cause misinterpretation of the results. It is necessary to use high-angle annular dark-field scanning TEM (HAADF STEM) to study their morphology as it is clearly visible in the example of TEM images of GSeNPs (Figure 3). When the image of simple TEM is studied, it can be concluded that large nanoparticles with dimensions of about 100 nm are obtained. However, when HAADF is applied, it can be seen that these are actually large clusters of small SeNPs with dimensions of approx. 12.15 nm. For comparison and to highlight the problem, a sample of GSeNPs was analysed using the DLS method. The results indicated particles with a size of 100 nm (Appendix A). This suggests that these clusters of smaller nanoparticles are stable enough that mixing during measurement does not break them up. Thus, already at the first stage of characterization of the obtained SeNPs, it is necessary to have a high-resolution microscope that will allow for the correct collection of data.

It should be recalled that from the point of view of potential medical applications of SeNPs, they must be smaller than 100 nm in order to cross cell membranes. In general, SeNPs obtained from the synthesis using green tea infusion were the biggest. When red tea was used, nanoparticles with average dimensions of 7.781 nm were obtained. Even smaller SeNPs were observed in the presence of black (4.891 nm) or white (3.938 nm) tea infusions. The exact particle size distribution is shown in the histograms in Appendix A. Spherical SeNPs are obtained only when green or black tea is used. In the case of red and white tea, nanorods are the products of the synthesis.

The size and shape of SeNPs are parameters that have a large impact on their antioxidant capacity. In our previous works, we postulated that the homogeneity of nanoparticles can also affect their properties [16]. The polydispersity index (PDI) is widely used to estimate the average uniformity of a particle solution. The higher the PDI value, the larger the particle size distribution in the analysed sample. PDI can also indicate nanoparticle aggregation along with the consistency and efficiency of particle surface modifications throughout the particle sample. A sample is considered monodisperse when the PDI value is less than 0.1 [48]. Based on the registered UV-Vis spectra, it was expected that the obtained SeNPs would be characterized by high polydispersity. This was indicated by several absorption maxima in the spectra, suggesting the presence of several fractions of SeNPs in the sample, differentiated in terms of size. The determined PDI values confirmed these assumptions. For all nanoparticles, the PDI values were greater than 0.1 and amounted to: 0.165 for GSeNPs, 0.298 for BSeNPs, 0.481 for RSeNPs and 0.381 for WSeNPs. Such a result does not preclude the use of the obtained SeNPs for biomedical purposes, as they meet the size criterion (they are smaller than 100 nm), but their polydispersity may translate into their antioxidant abilities.

The stability of the obtained SeNPs was investigated within a period of 48 h. The observed changes in the average diameter established by DLS analysis were not significant, indicating good stability of the investigated nanoparticles, as shown in Appendix A. Further studies involving the zeta potential are planned to be performed. The zeta potential of a nanoparticle is the measure of its surface charge density. It affects the stability of the nanoparticles, but also determines the way in which they interact with biological systems [49].

The typical energy dispersive spectroscopy (EDS) from SEM measurement was conducted to analyse the composition of obtained SeNPs. The obtained results are provided in Appendix A. Due to the difficulties in separating the obtained NPs, the measurement was carried out directly in the post-reaction mixture. The obtained spectra of all obtained SeNPs exhibit the characteristic signal of Se, C and O. Similar results of the EDS line scan spectrum indicating the nature of SeNPs were described in other studies [50,51]. In order to better study the structure and purity of the obtained SeNPs, a methodology for their isolation from the post-reaction mixture should be developed. The presented EDS studies only confirm the formation of SeNPs, but also emphasize the different composition of the sample matrix and, thus, how the types of tea used differ in terms of the content of individual elements. This composition has an undoubted influence on the course of the SeNP synthesis reaction.

### 4.4. Antioxidant Activity of Synthesized SeNPs

Selenium nanoparticles are attracting great attention due to their antioxidant and antibacterial properties [12,15]. As they can act in multiple ways in the human body, they are seen as substitutes for antibiotics [50]. Our results showed that all obtained nanoparticles showed lower tendency to scavenge DPPH radicals than tea extracts. These findings may be surprising as SeNPs are a known nanoantioxidant [51]. However, the polyphenolic compounds present in tea are largely responsible for the neutralization of DPPH radicals and these were used as selenium reducers during the synthesis of SeNPs, hence, their lower content in the post-reaction mixture and, thus, the lower ability of the sample to neutralize DPPH radicals. However, such a trend is not observed when analysing the results obtained for the neutralization of hydroxyl radicals. From the point of view of potential biomedical applications, the results of testing the ability to scavenge hydroxyl radicals are more reliable. These are presented in Table 2. The obtained selenium nanoparticles show a much higher ability to neutralize OH radicals in relation to the tea infusions from which they were obtained. The highest ability to neutralize free radicals among teas was shown by green tea (30.43 ± 0.92%), while for nanoparticles obtained with its use, this value is more than three-times higher (97.42 ± 1.3%). Such huge differences were observed for all teas. The total content of polyphenolic compounds determined for suspensions of selenium nanoparticles was significantly lower than for the corresponding tea infusions. However, this is completely understandable, as polyphenols are used in the synthesis of SeNPs, acting as reducers of selenium salts and, thus, their concentration decreases with the progress of the reaction. Reducing capacities were also determined for all samples via the CUPRAC method. As expected, higher values were obtained for tea infusions, which is again associated with a decrease in the content of polyphenolic compounds as a result of the course of the synthesis reaction. However, there is no correlation between the total content of polyphenolic compounds in the tea extract and the antioxidant capacity of the SeNPs obtained from them.

Nevertheless, there is a correlation between the shape of SeNPs and their ability to scavenge OH radicals. The most spherical BSeNPs showed the highest antioxidant activity, followed by GSeNPs (also spherical but bigger). The ability to neutralize OH radical was lowest for nanorods obtained from synthesis involving white and red teas. On the other hand, this correlation is not visible when the values of AOX indexes are examined. Here, BSeNPs, WSeNPs and GSeNPs showed a similar value for the AOX index. No correlation was observed between the size of nanoparticles and their ability to scavenge free radicals as well as the AOX index. This may, somehow, confirm our previous conclusion that not only the shape but also homogeneity is an important factor affecting the antioxidant ability of SeNPs [14]. It is difficult to determine which of the obtained SeNPs has the highest antioxidant capacity. Similar values of antioxidant indexes were obtained for BSeNPs, GSeNPs and WSeNPs. Only for RseNPs, a significantly lower value was obtained, which can be related to their polydispersity. The determined PDI value for RSeNPs was the highest (0.481), which suggests the lowest homogeneity of SeNPs from all the tested ones, and this parameter, as mentioned earlier, seems to have a large impact on the demonstrated antioxidant capacity. From the point of view of the ability to scavenge free radicals, DDPH assay is widely used in plant and food biochemistry to evaluate the free-radical scavenging effect of specific compounds or extracts. The reagent is commercially available and does not have to be generated before the assay as in other methods. However, from the point of view of living organisms, the ability to neutralize OH radicals is more important, as DPPH radicals do not occur in living organisms. Analysing the determining ability of SeNPs to neutralize hydroxyl radicals, clear correlation with the PDI value can be seen, indicating the homogeneity of the nanoparticles and their antioxidant activity. The higher the ability to neutralize OH radicals, the lower the PDI value. The exception is BSeNPs, which show a higher ability to neutralize OH radicals than the second in the order GSeNPs, although the PDI value determined for them (0.298) is higher than that obtained for GSeNPS (0.165). However, in this case, the criterion of size was decisive—GSeNPs are more than twice as large as BSeNPs. Based on our results, we are inclined to focus on the greater importance of determining the ability of SeNPs based on the ability to neutralize OH radicals than DPPH, which is additionally supported by the correlation of physical parameters of nanoparticles (shape, thickness, homogeneity) with the results obtained for this test.

### 4.5. Comparison of Properties of SeNPs Obtained Using Tea Extracts and by Chemical Synthesis

In our previous work, we obtained SeNPs via the chemical reduction of sodium selenate with ascorbic acid (AA) or cysteine (Cys) [16]. Comparing the results with those obtained from the SeNP synthesis using tea infusions, it should be stated that the green approach is unbeatable. The yield of the chemical reaction of the synthesis was 77% for the reaction with ascorbic acid and 88% with cysteine. In the case of syntheses using different types of tea, these values were higher than 90%. Moreover, SeNPs synthesized by green synthesis were much more stable than those obtained using AA or Cys, even with the addition of a stabilizer. Crucially, the effect of chemical synthesis was to obtain nanoparticles with the smallest dimensions, 80.0 ± 5.1 nm for Cys and 90 ± 7.3 nm for AA. In the case of synthesis using tea infusions, the biggest nanoparticles were about 12 nm for synthesis with green tea. All these things speak in favour of green synthesis. Its only drawback is the problem with the isolation of SeNPs from the post-reaction mixture. In this situation, other SeNP purification methods should be tried, e.g., dialysis.

## 5. Conclusions

The use of all analysed tea infusions allowed us to obtain SeNPs. From a dimensional point of view, all synthesized SeNPs have the potential to be used in medicine. Difficulties in their purification are a potential problem, and a thorough analysis of their parameters (size, shape) should be carried out on purified SeNPs, as the clean-up procedure has a great impact on the physical properties of SeNPs. The next step in the research on SeNPs synthesized by this method should be to develop a method for their purification or to consider the use of the post-reaction mixture as an oral mixture in a possible therapy with SeNPs.

## Figures and Tables

**Figure 1 antioxidants-11-02489-f001:**
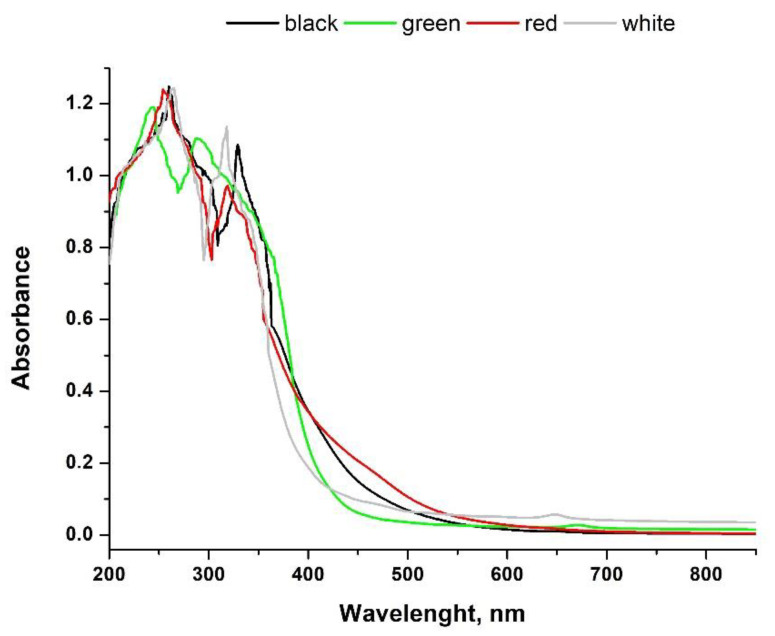
The UV-Vis spectra of examined teas: black, green, red and white.

**Figure 2 antioxidants-11-02489-f002:**
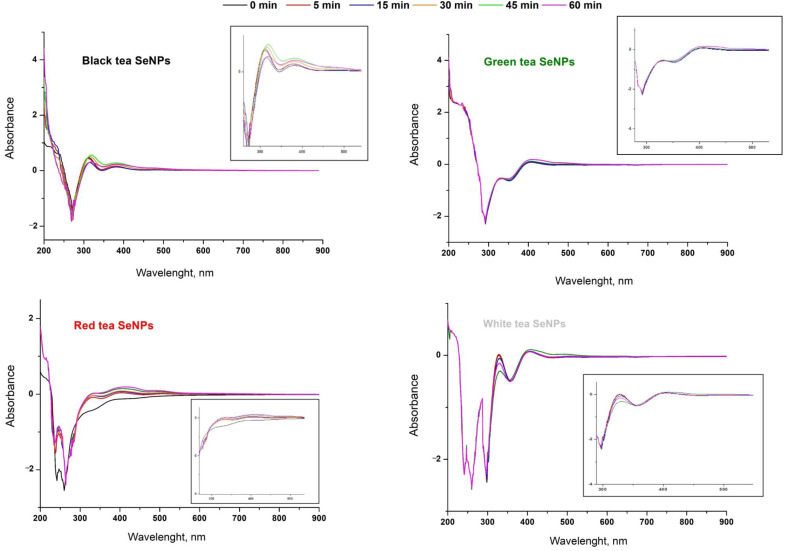
The UV-Vis spectra of selenium particles as a function of synthesis time. The box on each graph is an approximation of the 300–500 nm region.

**Figure 3 antioxidants-11-02489-f003:**
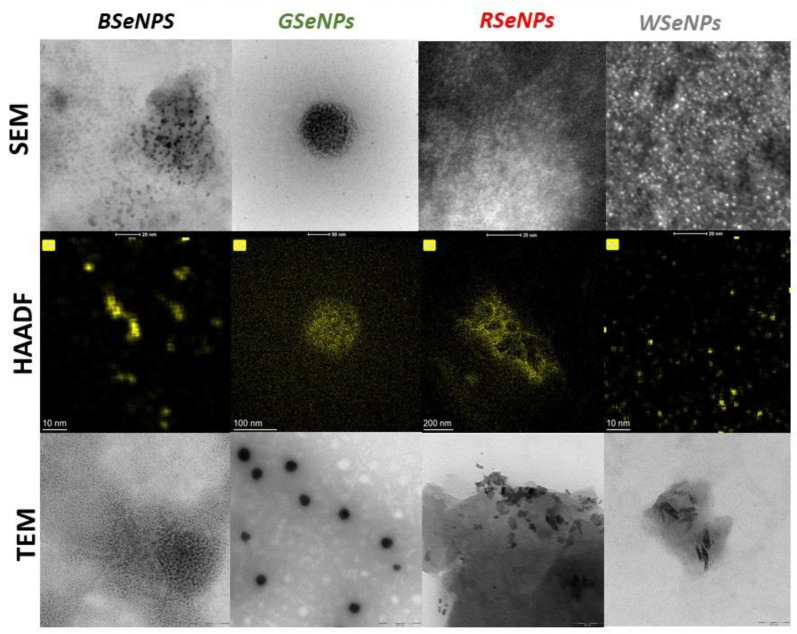
SEM, TEM and TEM (HAADF STEM) of obtained selenium nanoparticles. Abbreviations: BSeNPs—nanoparticles synthesized using black tea infusions, GSeNPs—green tea, RSeNPs—red tea and WSeNPs—white tea infusion.

**Table 1 antioxidants-11-02489-t001:** The content of polyphenolic compounds in tea infusions.

	Black Tea	Green Tea	Red Tea	White Tea
Flavonoids				
Catechin	4.76 ± 0.201 ^a^	11.7 ± 0.352 ^b^	6.21 ± 0.019 ^c^	20.3 ± 0.521 ^d^
Epicatechin	0.296 ± 0.010 ^a^	0.231 ± 0.007 ^b^	0.252 ± 0.010 ^c^	0.360 ± 0.014 ^d^
EGCG	30.1 ± 0.100 ^a^	70.0 ± 0.201 ^b^	6.92 ± 0.020 ^c^	64.0 ± 0.190 ^d^
Rutin	25.8 ± 0.772 ^a^	5.72 ± 0.171 ^b^	12.5 ± 0.375 ^c^	18.6 ± 0.56 ^d^
Polyphenolic Acids				
pHBA	38.3 ± 1.53 ^a^	7.92 ± 0.158 ^b^	17.8 ± 0.535 ^c^	13.5 ± 0.540 ^d^
Gallic acid	3.74 ± 0.140 ^a^	0.950 ± 0.030 ^b^	6.36 ± 0.190 ^c^	1.81 ± 0.054 ^d^
Chlorogenic acid	2.23 ± 0.090 ^a^	0.721 ± 0.022 ^b^	0.637 ± 0.019 ^c^	1.47 ± 0.044 ^d^
p-coumaric acid	4.47 ± 0.130 ^a^	1.72 ± 0.052 ^b^	7.16 ± 0.280 ^c^	4.85 ± 0.145 ^d^
Protocatechuic acid	0.378 ± 0.007 ^a^	0.177 ± 0.001 ^b^	0.300 ± 0.020 ^c^	0.224 ± 0.011 ^d^
Caffeic acid	0.103 ± 0.003 ^a^	<LOD ^2^	0.535 ± 0.010 ^b^	0.090 ± 0.002 ^a^

Data are expressed in mg L^−1^ of tea and as the means ± SD of three independent experiments. Different letters in each row indicate a difference at a significance level of *p* = 0.05. ^2^ LOD—limit of detection—the lowest concentration of the analyte that can be detected by applied method, calculated as 3 times signal to noise ratio.

**Table 2 antioxidants-11-02489-t002:** Antioxidant activity of tea infusions and selenium nanoparticles. Abbreviations: BSeNPs—nanoparticles synthesized using black tea infusions, GSeNPs—green tea, RSeNPs—red tea and WSeNPs—white tea infusion.

	*Antioxidant Capacity* *				
	FC [mg GA/L]	CUPRAC [mmol TrE/L]	DPPH [mmol TrE/L]	OH [%]	AOX Index
** *Tea Infusions* **					
**Black tea**	796.1 ± 2.25 ^a^	19.44 ± 0.70 ^a^	13.00 ± 0.30 ^a^	18.20 ± 0.07 ^a^	94.8 ± 2.3 ^a^
**Green tea**	1196 ± 3.78 ^b^	19.73 ± 0.60 ^a^	17.71 ± 0.40 ^b^	30.43 ± 0.92 ^b^	96.3 ± 2.8 ^b^
**Red tea**	848.7 ± 2.13 ^c^	14.70 ± 0.50 ^b^	12.31 ± 0.30 ^c^	24.25 ± 0.80 ^c^	97.8 ± 2.7 ^c^
**White tea**	1414 ± 4.21 ^d^	24.62 ± 0.91 ^c^	17.36 ± 0.51 ^d^	26.15 ± 0.79 ^d^	97.4 ± 3.1 ^c^
** *Selenium Nanoparticles* **					
**BSeNPs**	350.8 ± 1.61 ^e^	8.951 ± 0.27 ^d^	5.42 ± 0.19 ^e^	98.88 ± 1.2 ^e^	97.4 ± 2.5 ^c^
**GSeNPs**	176.9 ± 0.50 ^f^	9.080 ± 0.30 ^d^	5.33 ± 0.11 ^e^	97.42 ± 1.3 ^e^	97.1 ± 1.9 ^c^
**RSeNPs**	240.1 ± 0.87 ^g^	7.830 ± 0.30 ^e^	4.52 ± 0.17 ^f^	78.41 ± 1.5 ^f^	94.2 ± 2.1 ^a^
**WSeNPs**	598.8 ± 2.25 ^h^	10.33 ± 0.41 ^f^	6.72 ± 0.20 ^g^	80.43 ± 1.6 ^g^	97.3 ± 2.4 ^c^

* Data are expressed in mgGa/L, mmolTr/L or % and as the means ± SD of three independent experiments. Different letters in each column indicate a difference at a significance level of *p* = 0.05.

## Data Availability

The data presented in this study are available on request from the corresponding author.

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
