# Peer review of "Does the Type Matter? Verification of Different Tea Types’ Potential in the Synthesis of SeNPs"

_antioxidants, 2022, doi:10.3390/antiox11122489_

Round 1
Reviewer 1 Report
The manuscript titled "Does the type matter? Verification of different tea types' potential in the synthesis of SeNPs" is very interesting since it studies the green synthesis of the selenium nanoparticles, which present several benefits properties (antioxidants, antibacterial,...), being also less toxics than inorganic and organic selenium species. However, after reviewing this article, I have a series of comments for the consideration of the authors:
1. Nowadays, SeNPs have gained a lot of attention and are being widely studied in areas such as biomedicine, cancer therapy, and neurological diseases, but also as anti-inflammatory, anti-apoptotic, anti-bacterial, and antiviral agents. For this reason, the authors should emphasize the importance of nanoparticles in health and their biological applications (https://pubmed.ncbi.nlm.nih.gov/36157130/, https://pubmed.ncbi.nlm.nih.gov/35848122/ https://pubmed.ncbi.nlm.nih.gov/36079831/, https://pubmed.ncbi.nlm.nih.gov/33157197/ ). Likewise, it is necessary that the authors briefly expose the different types of synthesis of nanoparticles that exist (chemical, physical, or biological). In this way, it will be possible to better understand the importance of the green synthesis of SeNPs.
2. Authors should briefly describe all results in the "Results" section, and not show them solely through tables and figures.
3. The author should review the title of Table 2 and the figure caption in figure 3. The definition of the abbreviation RSeNPs is incorrect. Should it be red tea?
4. In page 7 line 243 there is a mistake. Revise this sentence please. Should be “from different types of tea, i.e. black, green, red and white”
5. It is not clear which are the most antioxidant SENPs: BSeNPs, GSeNPs, RSeNPs or WSeNPs? What is the best criterion to determine its antioxidant power? Why do WSeNPs show greater power to scavenge DPPH radicals than tea extracts (except green tea)? The value in the DPPH analysis is significant and lower than that obtained for the four types of tea extracts.
6. What does the percentage of OH indicate in Table 2? How is it determined? It is not clear either in the "Material and methods" or in the "Results" sections what it means and where that value is extracted from.
Author Response
Thank you for the valuable comments regarding our manuscript.
They were taking into consideration and the manuscript was filled. The added and corrected sentences are marked in red.
Detailed answers below.
Reviewer # 1
Nowadays, SeNPs have gained a lot of attention and are being widely studied in areas such as biomedicine, cancer therapy, and neurological diseases, but also as anti-inflammatory, anti-apoptotic, anti-bacterial, and antiviral agents. For this reason, the authors should emphasize the importance of nanoparticles in health and their biological applications (https://pubmed.ncbi.nlm.nih.gov/36157130/, https://pubmed.ncbi.nlm.nih.gov/35848122/ https://pubmed.ncbi.nlm.nih.gov/36079831/, https://pubmed.ncbi.nlm.nih.gov/33157197/ ). Likewise, it is necessary that the authors briefly expose the different types of synthesis of nanoparticles that exist (chemical, physical, or biological). In this way, it will be possible to better understand the importance of the green synthesis of SeNPs.
Thank you for your valuable suggestions. The introduction to the manuscript has been significantly extended and supplemented with the proposed literature.
- Authors should briefly describe all results in the "Results" section, and not show them solely through tables and figures.
We will keep this valuable point in mind when writing future papers. In the current version, we have not decided to limit the number of figures or tables so as not to disturb the structure of the manuscript. Perhaps this tendency is due to the fact that the two authors are visual learners and the results in the form of figures or tables are more readable for us.
- The author should review the title of Table 2 and the figure caption in figure 3. The definition of the abbreviation RSeNPs is incorrect. Should it be red tea?
This was corrected.
- In page 7 line 243 there is a mistake. Revise this sentence please. Should be “from different types of tea, i.e. black, green, red and white”
This sentence was revised.
- It is not clear which are the most antioxidant SENPs: BSeNPs, GSeNPs, RSeNPs or WSeNPs? What is the best criterion to determine its antioxidant power? Why do WSeNPs show greater power to scavenge DPPH radicals than tea extracts (except green tea)? The value in the DPPH analysis is significant and lower than that obtained for the four types of tea extracts.
The discussion regarding the antioxidant capacity of the obtained SeNps has been revised and extended.
- What does the percentage of OH indicate in Table 2? How is it determined? It is not clear either in the "Material and methods" or in the "Results" sections what it means and where that value is extracted from.
This was explained in Material and Method section.
Reviewer 2 Report
This article reports the Verification of different tea types' potential in the synthesis of SeNPs. The authors characterized the product, but the experimental data were less than adequate. Some issues should be improved before it be published. The detail comments are as follow.
1. When measuring DLS; please provide PDI and describe how the samples were suspended.
2. The authors are encouraged to provide stability evaluation of SeNPs.
3. There are some sentences in the manuscript that are not properly expressed or have grammatical problems. I hope the author will check and correct them carefully.
4. Please prove the purity of the SeNPs by performing an elemental analysis on it.
5. The authors should supplement the XRD analysis of the SeNPs.
6. The authors say that SeNPs are mainly used in biomedical applications, please add the biocompatibility of the product.
7. In this experiment, SeNPs were synthesized using a green method. Please compare how they differ in yield and performance from selenium nanoparticles synthesized by other conventional methods.
Author Response
Thank you for the valuable comments regarding our manuscript.
They were taking into consideration and the manuscript was filled. The added and corrected sentences are marked in red.
Detailed answers below.
Reviewer # 2
- When measuring DLS; please provide PDI and describe how the samples were suspended.
It was provided in Materials and Methods section as well as discussed in the lines 433-448.
- The authors are encouraged to provide stability evaluation of SeNPs.
It was added in lines 450-457. The corresponding graph has been added to the Supplementary Material section as Fig. 3S.
- There are some sentences in the manuscript that are not properly expressed or have grammatical problems. I hope the author will check and correct them carefully.
It as corrected. All language corrections are marked in red
- Please prove the purity of the SeNPs by performing an elemental analysis on it.
The data obtained from EDS measurements was added in Discussion section.
- The authors should supplement the XRD analysis of the SeNPs.
The results described in the paper are the first stage of research. We plan to extend it and then XRD analysis will be carried out.
- The authors say that SeNPs are mainly used in biomedical applications, please add the biocompatibility of the product.
The issue of the use of SeNPs in biomedical applications has been developed in the text, but the study of the biocompatibility of the obtained SeNPs on specific cell lines is planned in the future in cooperation with the Department of Biology of the University of Warsaw.
- In this experiment, SeNPs were synthesized using a green method. Please compare how they differ in yield and performance from selenium nanoparticles synthesized by other conventional methods.
A chapter on the comparison of SeNPs obtained by the methods described in this publication and our experience with the chemical method of synthesis has been added to the text.
Round 2
Reviewer 1 Report
My comment were adequately addressed. Thank you very much.
Reviewer 2 Report
The authors address my comments. This manuscript is recommended for consideration for publication. |